# Operation mode selection of NIMBY facility Public Private Partnership projects

**Hui Zhao, Jingqi Zhang**[ORCID]**\*, Yuanyuan Ge**

School of Management Engineering, Qingdao University of Technology, Qingdao, China

\* 459496030@qq.com

## Abstract

Construction of not in my back yard (NIMBY) facility Public Private Partnership (PPP) projects are feasible measures to realize sustainable urbanization. In order to ensure the smooth development of the NIMBY facility PPP projects, the problem of choosing the most suitable operation mode among many PPP modes is still scarce and unscientific. In order to select the operation mode of the PPP projects that best fits the characteristics of the project, this paper constructs the operation mode selection of the NIMBY facility PPP project. Firstly, the index system of operation mode selection of the NIMBY facility PPP project is determined. G1 subjective weighting method and information entropy objective weighting method are introduced to solve the optimal weight of each index. Grey correlation theory is used to improve TOPSIS method, and the calculation form of relative proximity degree is optimized to determine the most suitable operation mode for the project. In this paper, combined weighting and TOPSIS method are applied to the research of NIMBY facility PPP project, and the operation mode selection of NIMBY facility PPP project is established, which makes up the blank of this part. Finally, a PPP project in Qingdao, Shandong Province, China, is taken as an example to verify the applicability of the model. The effectiveness of this model was tested by comparing the results of TOPSIS method, Grey target model, Extended matter-element mode and GRA-TOPSIS. It is hoped to provide useful reference for the operation mode selection of NIMBY facility PPP project.

## 1. Introduction

In developed countries and regions with high economic level and good infrastructure, as well as developing countries and regions with underdeveloped economy and weak infrastructure, problems of urban not in my back yard (NIMBY) public facilities construction are common [1]. NIMBY facilities, as public facilities serving the general public in the region, generate benefits for the public and have negative externality effects, such as environmental pollution and threats to residents' lives and property. In addition, due to the large scale of investment, long profit cycle, and insufficient financial funds of relevant government departments, NIMBY facility projects cannot meet the strong demand for NIMBY facility construction in domestic cities, and it is difficult to meet the needs of urban development solely by relying on NIMBY facility projects [2]. In this context, in order to relieve the financial pressure of the government

**Data Availability Statement:** All relevant data are within the paper and its Supporting Information files.

**Funding:** The author(s) received no specific funding for this work.

**Competing interests:** The authors have declared that no competing interests exist.

and improve the efficiency of public goods supply, the Public Private Partnership (PPP) model is widely used in NIMBY facility projects. In this model, the public sector works with the private sector to provide public goods or services, leaving the government only responsible for regulation and the private sector to do the other cumbersome and complex work. This will not only solve the problem of government funding shortage and heavy workload, but also promote the diversification of investment entities. The government and the private sector can learn from each other, give play to their respective strengths and make up for each other's shortcomings. The two sides can develop mutually beneficial long-term goals and can provide high q1uality services to the public at the most effective cost. The practice of NIMBY facility PPP projects has proved that the operation mode of the project is the premise and key to the success of NIMBY facility PPP project [3]. Different from other PPP projects, adjacent to avoid many types of facilities of the PPP projects, involving multiple, complex construction system, overlapping, the boundary condition of each subsystem, and showed a larger difference, the main responsibility in liability attribution, risk allocation and income distribution is complex, reasonable PPP operation mode to a great extent, determines the adjacent construction and development situation of infrastructure projects. The PPP model encompasses a number of modes of operation, with some differences between them. For the NIMBY facility PPP project with long construction period and high risk, how to select the effective operation mode has become the key and difficult point for the successful implementation of such projects. Some scholars have focused on the influencing factors of PPP projects and analyzed the key evaluation factors for the operation mode selection of PPP projects. They believe that project dimension, government dimension, implementation dimension, risk dimension, income dimension and other factors are important indicators of PPP projects operation mode selection [4–7]. Scholars analyzed the definition of each operating mode PPP project, scope of application and trade framework, made the classification of the PPP project operation mode structure and choose the path [8]. Common multi-attribute decision making methods include RAFSI, MABAC, MAIRCA, VIKOR, etc. RAFSI (Ranking of Alternatives through Functional mapping of criterion sub-intervals into a Single Interval) could rank the Alternatives through Functional mapping of criterion sub-intervals into a Single Interval, which successfully eliminates the rank reversal problem [9]. MABAC (multi-attributive border approximation area comparison) model, which handles the complex and uncertain decision making issues by computing the distance between each alternative and the bored approximation area (BAA), has been investigated by an increasing number of researchers more recent years [10]. MAIRCA (multi attribute ideal real comparative analysis) can be used for quantitative and qualitative comprehensive analysis, with the advantages of objective evaluation, stable calculation results, and the calculation amount will not change greatly with the change of the number of variables [11]. Can VIKOR consider group utility maximization and individual regret minimization when decision-makers don't know how to express preferences [12]. The methods mentioned above can be used to build solutions and best ranking models A few scholars use different mathematical methods to select the PPP project operation mode, mainly concentrated in the improved TOPSIS method [13] and Monte carlo method [3]. In the existing literature, no scholars have applied TOPSIS method and information entropy to the PPP project of NIMBY facilities, but there have been relatively mature studies in other fields. Zolfani, (2020) [14] used VIKOR and TOPSIS, two famous multi-attribute decision making methods, to improve MADM. And the research study is focused on the analysis of the classical MADM methods based on logarithmic normalization. Durmić E, (2020) [15] combined Fucom method and Rough Saw model to solve the problem of operation mode selection, which provides a reference for the study of this paper. Dragan S (2020) [16] proposed the BWM (Best Worst Method) and the COPRAS (Compressed Proportional Assessment) models for the selection of the optimal off-road vehicle for the needs of the

SAF. Ramakrishnan (2020) [17] solves the problem of green supplier selection in the automotive industry by combining the cloud model with TOPSIS technology. Mohammad Noureddine (2019) [18] applied TOPSIS and MABAC (Multi-Attempted Border Approximation Comparison) to select the transport of Hazardous Material. Ivan Petrovic (2020) [19] has determined and evaluated the selection criteria for the radar position of ATC, and the hybridized Dematel-AHP-Topsis model was modified by using the interval type-2 Fuzzy sets (IT2FS), to ensure the radar in air traffic management to play a successful role. Xingle. Teng (2020) [20] through information entropy, grey correlation analysis method and TOPSIS method to determine the weight of indicators, judge the correlation degree and rank the decision units, and finally carry out a comprehensive evaluation of the development level of low-carbon economy in Shandong Province. Pengyu Dong (2020) [21] combined the grey relational analysis (GRA) and TOPSIS method, and used the method of AHP and information entropy to put forward a GRA-TOPSIS radiation source threat assessment model based on game theory. Hedrea, EL (2021) [22] presents the application of the tensor product (TP)-based model transformation approach to produce Tower CRrane (TCR) systems models. Through the analysis of previous scholars' articles, we can know that: GRA is a decision-making method to analyze "poor information" from a systematic point of view, which just overcomes the problems existing in the decision-making of TOPSIS method. GRA needs to determine the optimal value of each indicator, which is too subjective, and it is difficult to determine the optimal value of some indicators. TOPSIS method requires quantitative data, which is difficult and can only be used with more than two research objects. Therefore, the joint evaluation model of GRA and TOPSIS can not only consider the actual distance of the evaluated object in the multi-dimensional space, but also fully consider the correlation degree among various indicators. Entropy of information is a good description of the degree of chaos in a system, but it fails if the index changes very little or suddenly becomes larger or smaller.

Through the summary of the existing literature, it can be found that some scholars have studied the operation mode selection of PPP projects from different perspectives, and achieved certain results in the aspects of the influencing factors and selection methods of the operation mode. However, after combing the operation mode of existing PPP projects, the following prominent problems are found. (1) Although there have been a lot of PPP risk factors research results, some index systems contain too many evaluation indicators, without index simplification, and the evaluation of indicators is more complicated, leading to too much subjectivity in expert review, thus affecting the accuracy of evaluation data. (2) The selection of PPP project operation mode focuses on urban infrastructure PPP projects, transportation infrastructure PPP projects and urban rail PPP projects, etc., and there is a lack of research on the operation mode selection of NIMBY facility PPP projects. (3) In terms of existing research methods, more qualitative analysis and less quantitative analysis are used in the selection of operation methods of NIMBY facility PPP projects, which leads to too strong subjectivity of scheme selection and thus affects the scientific evaluation and selection results. At the same time, the mature selection and selection methods in other research fields are less applied in the operation mode selection of NIMBY facility PPP projects. Therefore, in order to ensure the success of the NIMBY facility PPP project, it is of great significance to conduct the preliminary feasibility study and the successful operation of the NIMBY facility PPP project.

This paper has the following practical and academic contributions. First of all, aiming at the problem that the index system of operation mode selection of NIMBY facility PPP project is not perfect, a set of operation mode selection index system specially applicable to the NIMBY facility PPP projects are constructed through Delphi method [23,24]. Secondly, in order to solve the problem that qualitative analysis is more than a quantitative analysis in current research methods, this paper proposes a methodological index system combining G1 method

and information entropy for combinatorial weighting. Thirdly, after comparison, this paper proposes a more suitable risk sharing method, that is, grey relational degree method (GRA) and improved TOPSIS method, optimizes the calculation form of relative proximity degree, and constructs the operation mode selection of NIMBY facility PPP project based on combination weighting and GRA-TOPSIS. Fourth, the information entropy and GRA-TOPSIS methods have been very mature in other fields, but the research on the PPP project of NIMBY facilities by these methods is still blank. In this paper, information entropy and GRA-TOPSIS are applied for the first time to the research on the operation mode of NIMBY facilities PPP project, making up the gap in this respect. Finally, the comprehensive ranking of the six operational modes is obtained through the calculation of the example, so as to determine the best operation mode of the NIMBY facility PPP project.

The rest of this study is organized as following. The second part introduces the research status of NIMBY facility PPP projects and the operation mode selection of PPP projects. In the third part, we construct a NIMBY facility PPP project operation mode selection index system. The fourth part constructs the operation mode selection of NIMBY facility PPP project based on combination weighting and GRA-TOPSIS. The fifth part verifies the feasibility and effectiveness of the operation mode selection through case analysis. Finally, the conclusion and the next step are given.

## 2. Literature review

### 2.1 NIMBY facility PPP projects

In the late 1970s, O 'Hare (1977) introduced the concept of NIMBY facilities into the academic community for the first time (Fig 1), while the analysis of NIMBY conflict as a specific form of urban local conflict by Dear and Taylor (1982) opened the prelude to a heated discussion on NIMBY phenomenon in the academic community [25]. In the 1980s and 1990s, scholars applied knowledge from different disciplines, such as politics, economics and psychology, to define the connotation of NIMBY, NIMBY facilities and NIMBY phenomenon. On the other hand, the causes and solutions of the NIMBY phenomenon are deeply discussed. It is found that the research on NIMBY phenomenon is also extended from the United States to Western countries such as Canada and Europe, and the research on NIMBY phenomenon is also increased in Asian countries. After entering the 21st century, more and more scholars proposed that NIMBY facility projects should vigorously adopt the PPP model, which provided a good guarantee for the development of NIMBY facility projects. Zhang X et.al [26] demonstrated the necessity of public participation in urban development by studying the dilemma and solutions of NIMBY facility governance, and proposed to combine NIMBY facility projects with PPP models to maximize the participation of public groups and balance the weight of public groups in decision-making process.Boyle.KG et.al [27] expounded the phenomenon of NIMBY conflict and pointed out the harm of NIMBY conflict. Taking the construction of a wind power plant as an example, they elaborated the views of stakeholders of the PPP model in detail and put forward constructive suggestions. Cheng Min et.al [28] combined the evolutionary game method with the system dynamics method to carry out the study. To understand the behavior choices of the public sector, the private sector and the public and their influencing factors in the NIMBY facility PPP projects, and take 19 NIMBY facility PPP projects in China as the research samples, the grounded theory method is adopted to analyze the risk of the NIMBY facility PPP projects. Top M et.al [29] used qualitative research methods to examine the extent to which public-private partnerships promoted the construction of NIMBY facility projects, and used descriptive analysis to analyze the qualitative research data and to analyze and evaluate the stakeholders of public-private partnership (PPP) model adopted in urban

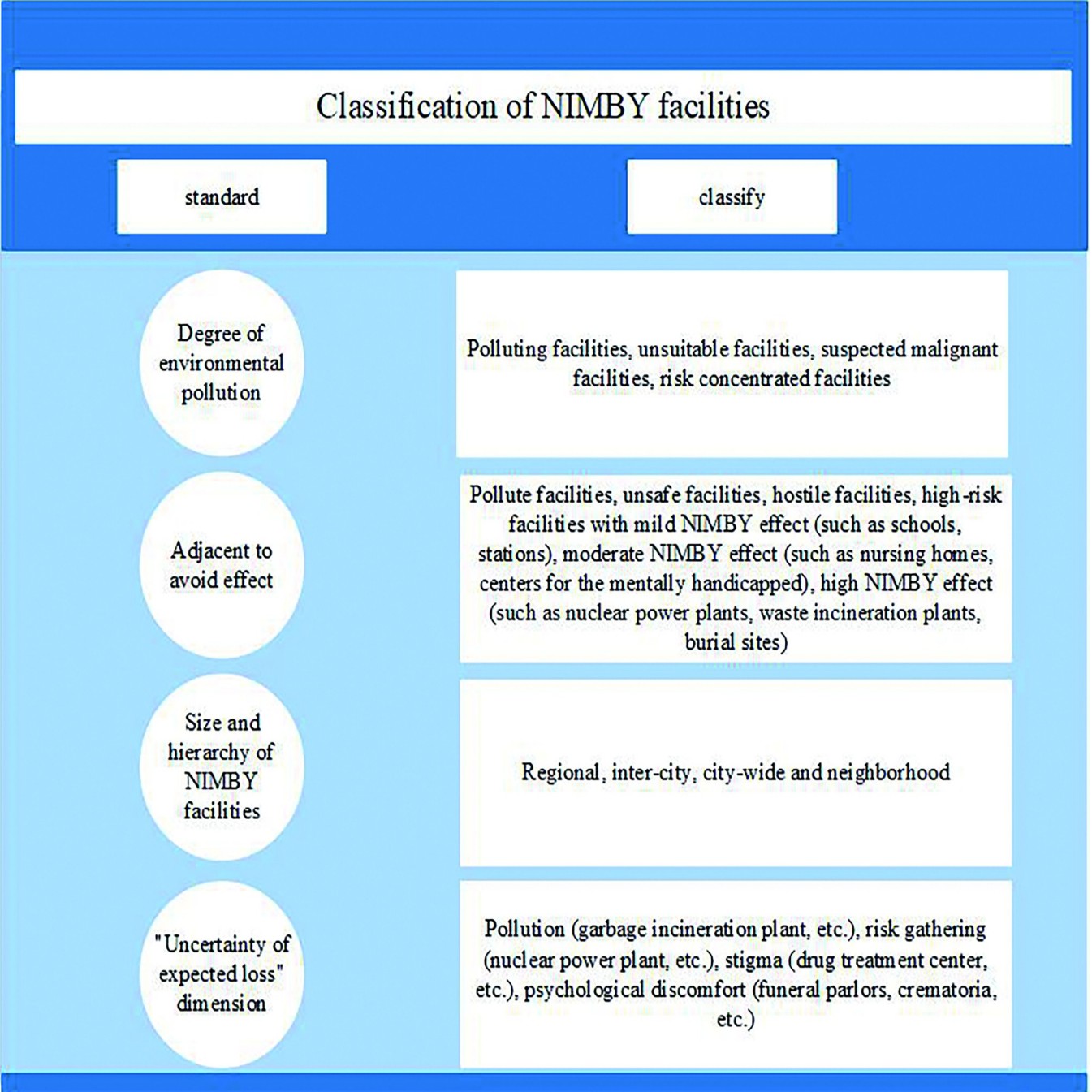

**Fig 1. Classification of NIMBY facilities.** The column on the left of the picture shows the criteria for the classification of NIMBY facilities. The column on the right of the picture is a detailed description of the classification of NIMBY facilities.

hospital construction projects in Turkey. Gharaee H et al [30] analyzed public-private partnership policy in primary health care (PHC), focusing on the experience of Iran's East Azerbaijan Province (EAP), and detailed the construction process of the NIMBY facility project. Through interviews with stakeholders and document analysis to collect data, it is recognized that the PPP model is a new successful experience of PHC in Iran. Support and development of this

policy can improve the quality and quantity of PHC and provide reference for other NIMBY facility projects to combine the PPP model. Ali A et.al [31] found the problem of illegal dumping of uncollected household waste in many major cities in East Africa and attributed it to the lack of public participation and poor implementation of existing legislation, recognizing that NIMBY facility projects require strong public-private partnerships, community participation and capacity-building. Azami-Aghdash S et.al [32], by listing the successful application of PPP NIMBY facilities in different countries, argued that PPP could be used as an option for road traffic injury (RTI) prevention, and used qualitative research and grounded theory to demonstrate the rationality of this idea. Tharun D et.al [33] claimed that public-private partnerships are increasingly being used in municipal solid waste (MSW) projects in India, and identified risk factors for MSW PPP projects in India by studying a comprehensive literature review, pointing out that future PPP projects of Nimby facilities need to pay special attention to these key risks. Geng S et.al [34] proposed a loan approval and evaluation framework for battery energy storage power station (BSPS) PPP for commercial banks to evaluate and select the optimal loan projects, so as to promote the development of BSPS-PPP projects and verify the rationality and effectiveness of PPP model in the construction of NIMBY facilities. Caiyun C et.al [35] studied the general situation, current situation, opportunities and challenges of public-private partnership in China's waste-to-energy incineration industry, and believed that the public-private partnership market was expanding rapidly, and a large number of well-funded and highly professional public-private partnership suppliers were increasing, but they were still second to the United States and Japan. Kirikkaleli D et.al [36] discussed the impact of renewable energy consumption and public-private investment in the energy sector on consumption-based carbon dioxide emissions in India. The study also suggested increasing public-private investment in renewable energy to achieve cleaner production processes, providing a basis for Nimby facility PPP projects. Baksh M et.al [37], referring to the implementation of the Core Cyber Security Plan, recognizes the need to strengthen public-private partnerships and provides guidance for further promoting the construction of Nimby facility PPP projects. In general, the NIMBY facility PPP project is in line with the current background of ecological civilization demand and is the main direction of government investment in both developed and developing countries. NIMBY facility PPP projects will continue to develop rapidly.

## 2.2 Operation mode selections of PPP projects

With the continuous improvement of the research, the operation mode of PPP projects presents diversified characteristics (in Table 1), more and more scholars have studied the operation mode selection of PPP projects. Mahdi IM et.al [3] think on the selection of the PPP project operation model, the influence of the factors mainly include the value, the characteristics of the project, interests and needs of the project cost, operation, and so on, to build the PPP project operation mode selection index system, and introduces the analytic hierarchy process (AHP) to compare and carries on the comparison to the PPP project operation model and measurement. Palaco I et.al [38] elaborates the advantages and disadvantages, operation mode and applicable conditions of PPP operation mode, and gives specific examples to demonstrate the application of PPP operation mode in urban infrastructure construction, by defining the scope of infrastructure projects and analyzing their characteristics. Guo Hualun et.al [39] established the corresponding selection index system, and summarized the key factors influencing the selection of PPP project operation mode, including investment target, quality target, schedule target, economic benefit, social benefit, operation efficiency, service level and financial expenditure. On this basis, combined with the analytic hierarchy process (AHP) principle, the PPP operation mode selection diagram is established. Delmon J et.al [40] studied

**Table 1. PPP operation mode.**

| Category | Type | Pattern | Contract period |
|---|---|---|---|
| Outsourcing class | Business Process Outsourcing | Service Outsourcing (SC) | 1 ~ 3 |
| | | Management Outsourcing (MC) | 3 ~ 5 |
| | Whole outsourcing | Design-Build (DB) | Indeterminacy |
| | | Design—Construction–Major-Maintenance (DBMM) | Indeterminacy |
| | | Design-Build-Operate (DBO) | 8 ~ 15 |
| | | Design-Build-Operate-Maintenance (DBOM) | Indeterminacy |
| | | Operations—Maintenance (O&M) | 5 ~ 8 |
| Franchise | BOT | Build-Own-Operate-Transfer (BOOT) | 25 ~30 |
| | | Construction—Leasing—Operation—Transfer (BLOT) | 25 ~30 |
| | TOT | Buy-Renew-Operate-Transfer (PUOT) | 8 ~ 15 |
| | | Lease-Renewal-Operate-Transfer (LUOT) | 8 ~ 15 |
| | PFI | Design-Build-Transfer-Operation (DBTO) | 20 ~25 |
| | | Design-Build-Invest-Operation (DBFO) | 20 ~25 |
| Private class | Full privatization | Build-Own-Operation (BOO) | Perpetual |
| | | BUY-RENEW-OPERATE (PUO) | Perpetual |
| | Some private | Stock right transfer | Perpetual |
| | | Joint venture to build | Perpetual |

First, the PPP operation mode is divided into three categories, and then into three types. Each type is then subdivided into several different forms, and the duration of the contract is listed.

some new market-oriented financing methods, and mainly analyzed the advantages, disadvantages and applicable conditions of BOT and TOT financing methods. On the basis of analyzing the characteristics of PPP project operation mode, Roghanian E et.al [41] proposed a model for selecting PPP project operation mode using QFD (Quality Function Deployment)-TOPSIS method. Yeo KT et.al [42] proposed that a reasonable and applicable operation mode will greatly affect the smooth implementation of PPP projects, and should be considered in the index in measuring and selecting the operation mode of PPP projects such as the project's own attributes, the government's ability and preference of the public sector, the private sector's ability and preference, and the degree of implementation of the relevant policies. Keating B et.al [43] analyzed the selection of PPP project design and operation mode from a qualitative point of view by comparing the Fremantle Port project in Australia with the Toll Road project in Indiana, USA. Yang et al [8] introduced the three-dimensional framework model of transaction and cooperation, pointed out the transaction and cooperation attributes of PPP mode, and analyzed the basic steps of selecting the PPP operation mode of public projects in detail on the basis of this theory. Yin Tailing [44] studied the selection path of PPP mode, analyzed the practical application of this path according to the specific characteristics of the project, and put forward four selection means. Ju-Yang Z et.al [45] introduced a variety of alternative models of PPP projects and made a simple analysis of their advantages and disadvantages, and proposed that in the reform of public hospitals in China, the government shares should adopt franchise mode or mixed ownership, and all property rights should be transferred to the government to ensure the preservation and proliferation of state-owned assets. Based on the project differentiation theory, Chen Jingwu ei.al [46] divides urban infrastructure projects into three categories, including quasi-operational infrastructure projects, non-operational infrastructure projects and operational infrastructure projects. Moreover, according to the specific characteristics and operation mechanism of PPP mode, appropriate operation modes are matched for different types of urban infrastructure PPP projects. Through literature research

and combined with the practical experience of PPP projects, Li Qian et.al [47] established the corresponding indicator system of mode selection according to the characteristics of existing urban rail transit PPP projects. The entropy weight grey target theory is introduced to build the PPP mode selection model of urban rail transit project stock, and the feasibility and practicability of the model are verified. Yunna W et.al [48] introduced risk assessment frameworks for seawater pumped storage projects in China under three typical public-private partnership management models, and introduced various PPP models to promote the construction of power plants. By summarizing the existing literature on the selection of PPP projects operation mode, it can be found that the existing research has the following two outstanding problems: (1) Although there are some achievements in the selection of PPP projects operation mode, the index system construction of PPP project operation mode selection is not systematic and comprehensive enough. (2) As for the selection model of PPP projects operation mode, the existing research focuses on the qualitative analysis, but the quantitative analysis is too few, which leads to too strong subjectivity of scheme selection, thus affecting the scientific nature of evaluation and selection results. Therefore, it is the focus of current and future research to construct a perfect index system and a model for the operation mode selection of NIMBY facility PPP projects.

## 3. Establishing operation mode evaluation index system of NIMBY facility PPP projects

In order to study the operation mode selection of NIMBY facility PPP projects, a measurement index system should be established. At present, PPP project index system is mainly realized through brainstorming method, Delphi method, scenario analysis method, etc. [23,49,50]. Given numerous PPP projects of NIMBY facilities at present, it is not difficult to gather experts and scholars in relevant fields. Therefore, this paper adopts the Delphi method to establish the index system of operation mode selection of NIMBY facility PPP project. Due to the large number of experts and scholars in relevant projects, the following aspects are taken into consideration: (1) experts should have at least 5 years of practical experience in NIMBY facility PPP projects. (2) the experts have participated in the training of NIMBY facility PPP projects and come from well-known universities. The number of experts is set to 8 by referring to a large number of relevant literature.

To set up scientific choice index system, according to the adjacent from the infrastructure PPP projects operation process and main operation characteristics, combined with China's other urban infrastructure PPP projects and mutual influencing factors from the infrastructure projects, and refer to the relevant literature, consulting the opinions of the expert inside course of study, the adjacent index to generalize from the infrastructure PPP projects integration, preliminary screening 30 adjacent from the infrastructure PPP projects operation mode selection factors. The specific steps are shown in Fig 2. PPP consulting institutions, government departments and engineering construction personnel were invited to score the rationality of the indicators in the form of survey vouchers. A total of 80 survey vouchers were issued and 67 valid questionnaires were received. The results of the questionnaire survey and expert feedback were summarized, and the reliability analysis was carried out with SPSS 24.0 software. The total Cronbach's α coefficient >0. 9 indicated that the reliability of the questionnaire was good, and the results of the questionnaire survey were true and reliable. Based on this, factor analysis is carried out to extract the principal component factors, eliminate the indicators with low correlation and cross repetition, reduce redundancy, thus reducing the influence of expert subjective uncertainty, and enhance the objectivity and representativeness of evaluation

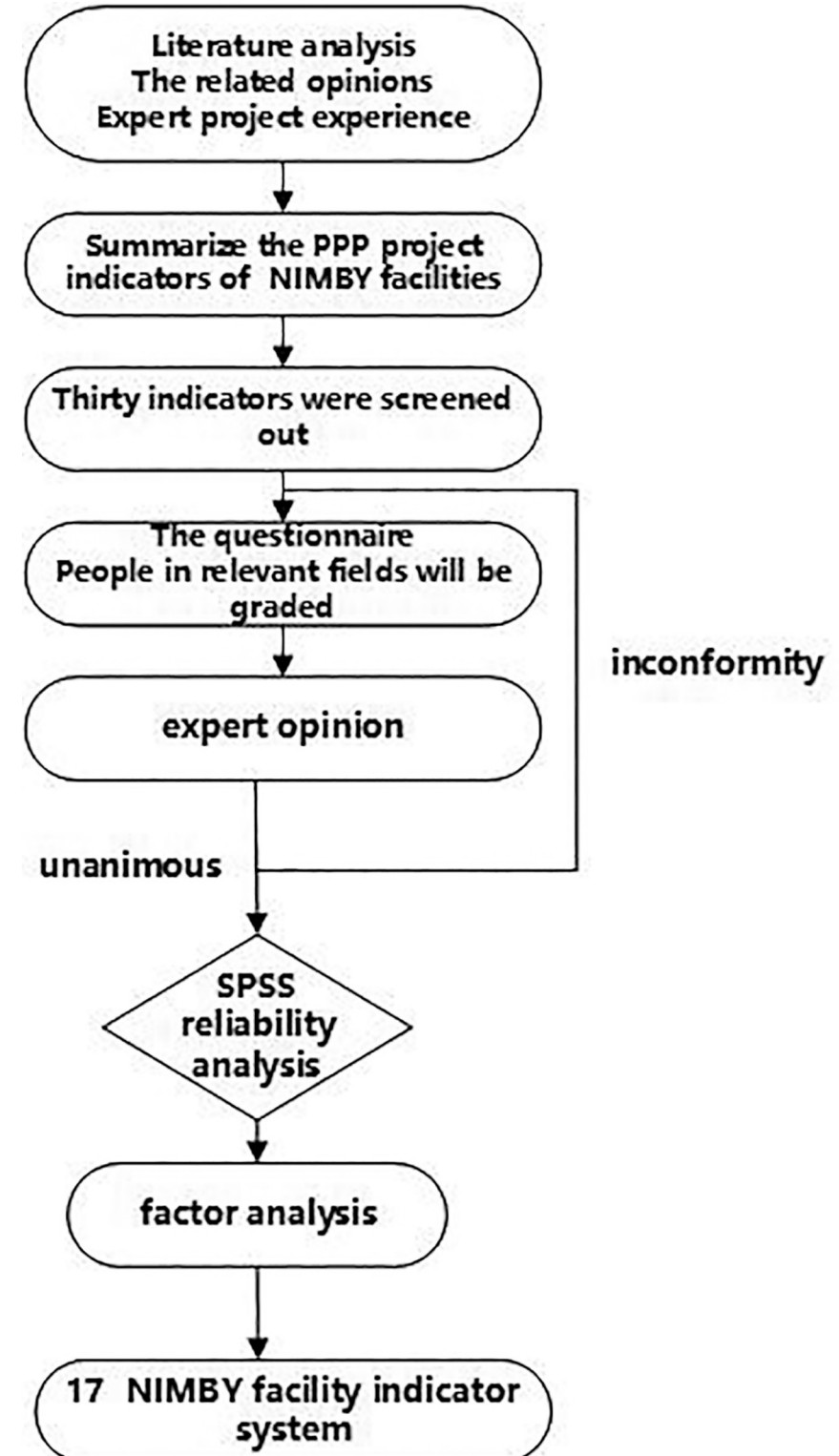

**Fig 2. Confirmation process of the index system.** The picture shows the confirmation process of the index system from top to bottom. It is worth noting that if the experts have different opinions on the 30 indicators, they will re-discuss and re-score them until they reach a consensus and proceed to the next step.

indicators. Through the above steps, the NIMBY facility PPP projects operation mode selection index system consisting of 17 indicators was finally established, as listed in Table 2.

## 4. Methodology

The operation mode selection of NIMBY facility PPP projects is a multi-attribute decision making problem. It is difficult for decision-makers to make accurate decisions through a single method. Then in this section, the grey relating-double benchmark method (GRA- TOPSIS) method is proposed to involve the following three stages. Firstly, brainstorming method, Delphi method and scenario analysis method were adopted to establish the evaluation index system of the operation mode of NIMBY facility PPP projects (shown in Section 3). Then, the weight of each evaluation index was determined by combining the subjective weighting method of G1 method and the objective weighting method of information entropy method. Finally, the GRA- TOPSIS method is used to obtain the optimal operation mode.

### 4.1 Determine the weight of each evaluation indicator

Measurement factors of operation mode have different influence on the PPP project of NIMBY facilities. It is very important to determine the specific weight of each index with appropriate methods before calculating the specific selection method of operation mode and after determining the index system of operation mode selection of NIMBY facility PPP project. It is very important to determine the weight, which will directly affect the accuracy of the operation mode selection in the multi-index decision. Although subjective weighting method can reflect the judgment of relevant experts on the importance of various factors, subjective weighting method is limited by the expert's personal knowledge and ability and relies too much on expert opinions. There are many commonly used subjective weighting methods, such as LBWA, FUCOM, BWM etc. The LBWA model is based on a pairwise comparison of the criteria through forming a non-decreasing array of the criteria importance levels. It enables rationally expressing the judgments of a decision-maker with a minimal number of comparisons [51]. Ahmad Naim (2020) [52] compared FUCOM and BWM and concluded that BWM has less pair-wise Verbs and leads to FUCOM more reliable solution. The author claimed that FUCOM offers better consistency, reduced pair-wise comparisons, and flexibility of measurement scale. Different from the above methods, G1 method does not need consistency test and is more convenient to operate. Therefore, this paper chooses G1 method to calculate the weight.

The objective weighting method can make use of the nature of the effective data, but the objective weighting method relies on statistical and mathematical methods and ignores the human factors in the decision-making process, which makes it distinct from the display.

There are many commonly used objective weighting methods, such as CRITIC, FANMA, information entropy, etc. Milićević (2011) [53] proposed the information entropy method, the CRITIC method and the Fanma method, and compared and analyzed the three methods. CRITIC considers the size of the variability of indicators while taking into account the correlation between indicators, not that the greater the number is, the more important it is, and makes full use of the objective attributes of the data itself for scientific evaluation. Fanma is objective and generates weights of criteria directly from the criterion value variants. It also eliminates the problem of subjectivity and incompetence. Compared with other objective weighting methods, the use of information entropy is not limited by its scope. It can be used in any evaluation problem, and it can also eliminate the indexes that do not contribute much to the evaluation results in the index system. In view of the shortcomings of subjective and objective weighting methods, this paper adopts the combined weighting method (combining the G1

**Table 2. Operation method measurement standard system of NIMBY facility PPP projects.**

| Target layer | Criterion layer | Index level | Guidelines |
|---|---|---|---|
| Selection of operation mode of NIMBY facility PPP projects | Project Features B1 | Project Construction Category C1 | "Project Construction Category" refers to the major construction types of PPP projects [3] |
| | | Financing Scale C2 | "Financing Scale" refers to the amount of financing required for PPP projects [38] |
| | | Project market demand C3 | "Project Market Demand" refers to the market demand for this type of PPP projects [3] |
| | Government dimension B2 | Government PPP Inherent Experience C4 | "Government's inherent PPP Experience" refers to the amount of experience the government has in guiding the construction of PPP projects [29] |
| | | Government policy leans towards C5 | "Government policy inclination" refers to the degree of government policy inclination to use PPP mode for financing [29] |
| | | Government fiscal capacity C6 | "Government fiscal capacity" refers to the size of the government's own fiscal capacity [29] |
| | | Government regulatory capability C7 | "Government regulatory capacity" refers to the extent to which the government supervises PPP projects [29] |
| | Implement dimension B3 | Financing process complexity C8 | The term "financing process complexity" refers to the complexity of the financing process by using different PPP Operations [38] |
| | | Technical adaptability C9 | The term "technical adaptability" refers to the degree of technical adaptability to different PPP operations [39] |
| | | Complexity of property rights change C10 | The term "complexity of title changes" refers to the complexity of title changes at the end of PPP projects [38] |
| | Revenue dimension B4 | Return on investment level C11 | The term "ROI level" refers to the level of return of a PPP projects during its operating life [39] |
| | | Fee pricing mechanism C12 | The term "fee-pricing mechanism" refers to the reasonableness of the fee-pricing mechanism for PPP projects in different operating modes [39] |
| | | Investment payback period C13 | The term "payback period" refers to the applicability of different operational payback periods to sewage treatment projects [39] |
| | Risk dimension B5 | Financial risk C14 | "Financial Risk" refers to the loss of PPP projects due to changes in the financial market [38] |
| | | Construction risk C15 | "Construction Risk" refers to the loss incurred during the construction of PPP projects [3] |
| | | Political risk C16 | "Political risk" refers to the loss of PPP projects due to adverse effects from policies [38] |
| | | Force Majeure Risk C17 | "Force Majeure Risk" refers to the loss of PPP projects caused by force major factors [38] |

First, Selection of operation mode of NIMBY facility PPP projects are divided into five first-level indicators. They were then divided into 17 secondary indicators and explained in detail.

method and information entropy) Yu-shan H et al [54] to determine the weight of the measurement factors of each operation mode. Compared with the analytic hierarchy process, the G1 method does not need to construct judgment matrix, nor does it need to carry out consistency test. G1 method is more concise and can better reflect the subjective preferences of experts. However, this kind of weight method relies too much on the score of experts and cannot reflect the change of objective conditions. The entropy of information is mainly determined by the entropy of each attribute index through all alternative schemes. The larger the entropy is, the higher the information disorder is, and the smaller the contribution of this index to the measurement results [55]. The weight value obtained by experts' scoring is often highly subjective, and the results obtained by different experts' scoring also have a certain potential deviation. The theory of information entropy greatly reduces the subjectivity of the score and makes the weight more real and reliable. Therefore, only combining the two

methods can effectively avoid subjective and objective bias and make the evaluation process more rigorous and reasonable.

**4.1.1 Subjective weight using G1 method.** Professor Guo Yajun put forward an improved subjective weight method—G1 method on the basis of AHP theory [56]. In the G1 method, the measurement indicators are sorted according to their importance by referring to expert opinions, and then the adjacent measurement indicators are judged, and on this basis, the quantitative assignment is carried out. This not only fully reflects the opinions of experts, but the ranking of importance is not arbitrary. The steps of determining index weight of G1 method are as follows.

Step 1. Determine the order relation of each index: Experts rank each index according to the importance of each index to the target layer. Assume that there are n measurement indicators that constitute the index set $C = \{c_1, c_2 \ldots, c_n\}$, if the index $c_l$ ($l \in 1, 2 \ldots, n$) With respect to the index $c_k$ ($k \in 1, 2 \ldots, n$), if the important degree is higher, it is denoted as $c_l > c_k$, and so on, so as to obtain the importance ranking of all measurement indicators.

Step 2. Determine the ratio of importance between adjacent indicators. By comparing the importance of $x_j$ And $x_{j+1}$, it is concluded that $r_j = c_j/c_{j+1}$ (j = 1, 2 \ldots, n-1), the value of $r_j$ Is shown in Table 3.

Step 3. Calculate the weight of the index relative to the criterion layer. According to the following formula, the weight value of the index j under the G1 weighting method to the criterion layer is obtained:

$$\begin{cases} u'_j = \left(1 + \sum_{j=1}^{n-1} \prod_{i-j}^{n-1} r_i\right)^{-1} \\ u'_{j+1} = r_j u'_j (j = 1, 2, \ldots, n-1) \end{cases} \tag{1}$$

Step 4. Calculate the weight of the metric relative to the target layer. According to the following formula, the weight value of the index j under the G1 weighting method to the target layer is obtained:

$$\omega'_j = u'_j \times u'_d \tag{2}$$

Among them, $u'd(d = 1,2,\ldots)$ represents the weight value of the criterion layer of item D, and the solution process is similar to the weight solution of the index layer.

**4.1.2 Objective weights based on the entropy.** In 1948, Shannon put forward the concept of information entropy based on the theory of thermal entropy in physics and information theory, which is used to describe the average uncertainty between signals [57]. That is, if a system Y may have several different states, $y_1, y_2 \ldots, y_n$, The probability of $y_i$ (i = 1,2 \ldots, n) Occurring is $\omega_i$, then the information entropy of the system is:

$$H(Y) = -\sum_{i=1}^{n} p_i \log_2 p_i \ \ (0 \leq p_i \leq 1, \sum_{i=1}^{n} p_i = 1) \tag{3}$$

The specific steps of determining the weight of each indicator of the entropy are as follows.

**Table 3. The relative importance of a standard.**

| $r_j$ Value | Meaning |
|---|---|
| 1.0 | Indicators $c_j$ are as important as indicators $c_{j+1}$ |
| 1.2 | The index $c_j$ is slightly more important than the index $c_{j+1}$ |
| 1.4 | The index $c_j$ is obviously more important than the index $c_{j+1}$ |
| 1.6 | The index $c_j$ is very important relative to the index $c_{j+1}$ |
| 1.8 | The index $c_j$ is extremely important relative to the index $c_{j+1}$ |

The scoring range and the scoring standard of the index are stipulated.

Step 1. Building the raw data matrix. Assuming that there are m alternative scheme items and n measurement indicators, the original data matrix is expressed as $A = (x_{ij})_{m \times n}$

$$A = \begin{bmatrix} X_{11} & X_{12} & \dots & X_{1n} \\ X_{21} & X_{22} & \dots & X_{2n} \\ \vdots & \vdots & \ddots & \vdots \\ X_{m1} & X_{m2} & \dots & X_{mn} \end{bmatrix} \tag{4}$$

Among them, $x_{ij}$ ($j = 1,2,\dots,m; j = 1, 2,\dots,n$)represents the evaluation value of scheme $i$ under j indicators. $A_{ij}$ is expressed as the observed value of index $j$ in scheme $i$.

Step 2. The characteristic proportion of scheme $i$ under the $j$ index was calculated:

$$p_{ij} = X_{ij} / \sum_{i=1}^{m} X_{ij} \tag{5}$$

Step 3.Entropy value of index j is calculated as following:

$$E_j = -k \sum_{i=1}^{m} p_{ij} \ln p_{ij}, \text{ Among them, } k = 1/l_{mn} \tag{6}$$

Step 4. Calculate the weight of the index $j$:

$$\omega_j^{''} = \frac{1 - E_j}{\sum_{j=1}^{n} (1 - E_j)} \tag{7}$$

**4.1.3 The combined weight of G1 method and information entropy.** The combined weight method combining G1 method and information entropy combines the professional opinions of experts in G1 subjective weight weighting method, and uses the objective weight method of information entropy to avoid the subjective error, making the final index weight more scientific and reasonable. The key to this method is to determine the proportion of weight $\omega_j'$ of G1 and weight $\omega_j''$ of information entropy in the comprehensive weight $\omega_j$:

$$\text{Step 1}: w_j = v_1 w_j' + v_2 w_j'' \ (j = 1, 2, \dots, n; \ v_1 > 0, \ v_2 > 0, \ v_1^2 + v_2^2 = 1) \tag{8}$$

Among them, $v_1$, $v_2$ represent the undetermined coefficients of subjective and objective weights. The solution of the undetermined coefficient can be transformed into the following optimization problem.

$$\text{Step 2}: \begin{cases} maxF(v_1, v_2) = \sum_{i=1}^{m} \left( \sum_{j=1}^{n} (v_1 w_j' + v_2 w_j'') \right) \\ s.t. \ v_1^2 + v_2^2 = 1 \quad v_1, v_2 \geq 0 \end{cases} \tag{9}$$

Step 3: According to the Lagrange extremum condition, we can get:

$$\begin{cases} v'_1 = \dfrac{\sum_{i=1}^{m} \sum_{j=1}^{n} \omega'_j x_{ij}}{\sqrt{\left(\sum_{i=1}^{m} \sum_{j=1}^{n} \omega'_j x_{ij}\right)^2 + \left(\sum_{i=1}^{m} \sum_{j=1}^{n} \omega''_j x_{ij}\right)^2}} \\[4ex] v'_2 = \dfrac{\sum_{i=1}^{m} \sum_{j=1}^{n} \omega''_j x_{ij}}{\sqrt{\left(\sum_{i=1}^{m} \sum_{j=1}^{n} \omega'_j x_{ij}\right)^2 + \left(\sum_{i=1}^{m} \sum_{j=1}^{n} \omega''_j x_{ij}\right)^2}} \end{cases} \tag{10}$$

Step 4: Normalization of $v'_1$ and $v'_2$ can be obtained as follows:

$$\begin{cases} v_1 = \dfrac{v'_1}{v'_1 + v'_2} \\[2ex] v_2 = \dfrac{v'_2}{v'_1 + v'_2} \end{cases} \tag{11}$$

## 4.2 Construction of operation mode selection model of NIMBY facility PPP project

Due to the complexity of the operation mode selection model of NIMBY facility PPP projects, scholars no longer use a single method to construct the model. In this context, a few scholars use the grey relational degree method to construct the model. The basic idea of grey relational analysis (GRA) is to determine the correlation degree between the comparison sequence set and the reference sequence according to the geometric similarity degree between the curve family formed by each comparison sequence set and the curve family formed by the reference sequence [58]. The greater the correlation between the two, the more consistent the trend between the curves will be. However, the existing evaluation results do not consider the correlation between the evaluation object and the negative ideal solution and are mostly one-sided. TOPSIS method is a kind of ordering method that approximates the ideal solution by setting the ideal solution and the negative ideal solution. The basic idea of TOPSIS is to evaluate each object based on the two benchmarks that are close to or far away from the ideal solution and the negative ideal solution, and then to sort [59]. However, taking proximity as a measurement standard can not reflect the future development trend of data series, but only show the position relationship between curves. If the distance between the evaluation object and the ideal solution is similar under the condition that the index values differ greatly, the similar result will still be obtained. Therefore, the above two methods have their own advantages and disadvantages. In the case of limited data of NIMBY facility PPP projects, the Grey Relation Double Benchmark Method (GRA-TOPSIS) [20] is formed by the effective combination of the two methods to make full use of the advantages of the two methods to ensure the accuracy of judgment from the perspectives of position and shape similarity. The calculation steps are as follows.

Step 1: Multiply the combined weight vector ω calculated by the G1 method and information entropy with the normalized matrix $\bar{F}$ to obtain the weighted standardized matrix V, and determine the positive ideal plan $Z_j^+$ and negative ideal plan $Z_j^-$ in the risk assessment of the

NIMBY facility PPP project to be evaluated as follows:

$$Z = (z_{ij})_{txz} = (\omega_j y_{ij})_{uz}$$
$$z_j^+ = \max_i z_{ij} | z_{ij} \in Z^+, \min_i z_{ij} | z_{ij} \in Z^- \tag{12}$$
$$= z_1^+, z_2^+, \cdots, z_z^+$$

$$z_j^- = \min_i z_{ij} | z_{ij} \in Z^+, \max_i z_{ij} | z_{ij} \in Z^- \tag{13}$$
$$= z_1^-, z_2^-, \cdots, z_z^-$$

In the formula, $Z_j^+$ represents that the larger the index value is, the better the index; $Z_j^-$ represents that the smaller the index value is, the better the index.

Step 2: Calculate the grey correlation coefficient $\xi_{ij}^+$ and $\xi_{ij}^-$ of the i index and the j scheme with positive ideal scheme $z_j^+$ and negative ideal scheme $Z_j^-$ respectively.

$$\xi_{ij}^+ = \frac{1}{z} \sum_{j=1}^{n} \frac{\min_i \min_j |z_{ij} - z_j^+| + \rho \max_i \max_j |z_{ij} - z_j^+|}{|z_{ij} - z_j^+| + \rho \max_i \max_j |z_{ij} - z_j^+|} \tag{14}$$

$$\xi_{ij}^- = \frac{1}{z} \sum_{j=1}^{n} \frac{\min_i \min_j |z_{ij} - z_j^-| + \rho \max_i \max_j |z_{ij} - z_j^-|}{|z_{ij} - z_j^-| + \rho \max_i \max_j |z_{ij} - z_j^-|} \tag{15}$$

In the formula, $\rho$ is the coefficient of resolution and $\rho = 0.5$ according to experience.

Step 4: Calculate Euclidean distances $d_i^+$ and $d_i^-$ between each evaluation scheme and positive and negative ideal schemes

$$d_i^+ = \sqrt{\sum_{j=1}^{2} (z_{ij} - z_j^+)^2}$$
$$d_i^- = \sqrt{\sum_{j=1}^{2} (z_{ij} - z_j^-)^2} \tag{16}$$

Step 5: Standardize the distance and correlation degree between each scheme and positive and negative ideal schemes

$$R_i^+ = \frac{\gamma_i^+}{\max \gamma_i^+}, R_i^- = \frac{\gamma_i^-}{\max \gamma_i^-}$$
$$D_i^+ = \frac{d_i^+}{\max d_i^+}, D_i^- = \frac{d_i^-}{\max d_i^-} \tag{17}$$

According to the definition of grey correlation, the larger the value of $R_i^+$, the higher the correlation degree with the positive ideal solution; otherwise, the lower the correlation degree with the positive ideal solution; similarly, the larger the value of $R_i^-$, the higher the correlation degree with the negative ideal solution; According to the definition of Euclidean distance, the larger the value of $D_i^+$, the farther the distance between the scheme and the positive ideal scheme, and the lower the degree of closeness; otherwise, the closer the distance between the scheme and the positive ideal scheme, and the higher the degree of closeness; similarly, the larger the value of $D_i^-$, the farther the distance between the scheme and the negative ideal scheme, and the lower the degree of closeness [60]. An improved formula for calculating the relative closeness degree is established.

Step 6: Calculate the relative closeness degree $\pi_i$ between each alternative scheme and the combination of positive ideal scheme $Z_j^+$ and negative ideal scheme $Z_j^-$.

$$\pi_i = \frac{aR_i^+ + bD_i^-}{(aR_i^* + bD_i^-) + (aR_i^- + bD_i^+)} \tag{18}$$

In the formula, both a and b belong to [0, 1], and a + b = 1. Normally, a = b = 0.5.

Rank the relative closeness value $\pi_i$ obtained from the largest to the smallest, and the result is the ranking of each scheme. The scheme with the largest relative closeness value $\pi_i$ is the optimal scheme.

## 5. Case study

Currently, there are a great number of NIMBY facilities projects all over the world, and more and more projects choose to adopt the PPP model. Although there are differences among these NIMBY facility PPP projects in terms of scale, location and complexity, the operational logic relationship is basically the same. Among them, the PPP project of integrated waste treatment in Qingdao, Shandong Province, China, is the most representative one at present. This paper takes the PPP project of Qingdao City, Shandong Province, China as an example to conduct an empirical study to verify the feasibility and effectiveness of the operation mode selection. Shandong Province is one of the provinces with the highest population concentration in China, so there are many NIMBY problems. Qingdao plays a pivotal role in the implementation of the NIMBY facility PPP project in Shandong Province. At the same time, the relevant research on the integrated waste treatment project of Qingdao needs to be strengthened. The total investment of this project is estimated to be about 137.07 million yuan. The construction period of the first phase of the project is 2 years and the operation period is 28 years. The Project Company shall be responsible for the whole process of financing and investment, construction and implementation, operation and maintenance management and asset management of the project, and shall be responsible for its own operation and profit and loss. The income of MSW project is based on the income of on-grid electricity fee, slag income and government feasibility gap subsidy. Waste incineration power generation, electricity costs, slag income as a supplement, insufficient part of the government feasibility gap subsidies to subsidize. The PPP Project Contract clearly stipulates that the implementation agency has the right to supervise the whole process, and the project company will hand over the project to the government or an agency designated by the government free of charge after the expiration of the cooperation period [61]. Combined with the urban characteristics of Qingdao, the following six PPP operation modes will be selected to construct the comprehensive waste treatment PPP project in Qingdao: DBO, TOT, BOO, DBFO, BOT and equity transfer.

### 5.1 The index system is weighted based on G1—information entropy combination weighting method

Step 1. Rank and score the indicators according to their importance

$r_j$ = 1.8,1.8,1.4,1.4,1.6,1.2,1.4,1.2,1.4,1.2,1.4,1,1.2,1.4,1.2,1.2,1.

Step 2. Subjective weight can be calculated according to Formula (1) and (2)

$\omega_j'$ = 0.037,0.059,0.112,0.052,0.047,0.062,0.048,0.033,0.048,0.051,0.094,0.079,0.034,0.058,0.081,0.074,0.063.

Step 3. The information entropy is obtained by standardizing the data

$E_j =$

0.9146,0.9697,0.8720,0.8313,0.7616,0.7152,0.7428,0.8530,0.8243,0.8172,0.8276,0.8749,0.8392,-0.9426,0.8263,0.9163,0.8482.

Step 4. The objective weight is obtained by formula (7)

$\omega_j^{''} =$

0.045,0.047,0.071,0.059,0.038,0.057,0.060,0.038,0.051,0.058,0.084,0.087,0.050,0.053,0.072,0.078,0.076.

Step 5. Substitute $\omega_j'$ and $\omega_j^{''}$ into formula (9) and (10)

$v_1' = 0.42, v_2' = -0.84$

Step 6. Substitute $v_1'$ and $v_2'$ into formula (11)

$v_1 = 0.5, v_2 = 0.5$

Step 7. Substitute $v_1$ and $v_2$ into formula (8)

$w_j =$

0.041,0.047,0.071,0.059,0.038,0.057,0.060,0.038,0.051,0.058,0.084,0.087,0.050,0.053,0.072,0.078,0.076.

Step 8. The subjective weight, objective weight and comprehensive weight of each index can be calculated according to the formula of combination weighting, as showed in Table 4.

## 5.2 Evaluate with the GRA-TOPSIS method

Step 1. Firstly, the weighted normalized decision matrix Z can be obtained according to formula (9)

$$
Z = \begin{pmatrix}
0.2500 & 0.5000 & 1.0000 & 1.0000 & 0.0000 & 0.5000 & 1.0000 & 1.0000 \\
0.0000 & 0.2500 & 0.6667 & 0.6667 & 0.2000 & 0.4000 & 0.5000 & 0.0000 \\
0.5000 & 0.0000 & 1.0000 & 0.4000 & 0.2500 & 0.0000 & 0.0000 & 0.0000 \\
1.0000 & 1.0000 & 1.0000 & 1.0000 & 0.3333 & 0.2000 & 0.6667 & 0.5000 \\
0.5000 & 0.7500 & 0.5000 & 0.0000 & 1.0000 & 1.0000 & 0.5000 & 0.0000 \\
0.0000 & 0.2500 & 0.3333 & 0.6667 & 0.2500 & 0.6667 & 0.0000 & 0.6667 \\
0.5000 & 0.6667 & 0.5000 & 1.0000 & 0.0000 & 0.6667 & 1.0000 & 0.0000 \\
0.3333 & 0.3333 & 0.6667 & 0.0000 & 1.0000 & 0.0000 & 0.0000 & 0.0000 \\
1.0000 & 0.5000 & 1.0000 & 0.5000 & 1.0000 & 0.5000 & 0.3333 & 0.5000 \\
0.5000 & 0.0000 & 0.4000 & 0.0000 & 0.0000 & 1.0000 & 0.5000 & 0.6667 \\
1.0000 & 0.6000 & 0.3333 & 0.2500 & 1.0000 & 0.4000 & 0.0000 & 0.0000 \\
0.3333 & 0.6667 & 0.6000 & 0.0000 & 0.2000 & 0.0000 & 0.5000 & 0.0000 \\
0.0000 & 0.7500 & 0.3333 & 0.0000 & 1.0000 & 1.0000 & 0.6000 & 0.2000 \\
1.0000 & 0.3333 & 1.0000 & 0.6667 & 0.6667 & 1.0000 & 0.0000 & 0.4000 \\
0.5000 & 0.3333 & 0.0000 & 0.5000 & 0.0000 & 0.6667 & 0.2500 & 0.0000 \\
0.0000 & 0.0000 & 1.0000 & 0.4000 & 0.0000 & 0.7500 & 0.6000 & 0.6000
\end{pmatrix}
$$

Step 2. Weighted normalization matrix Z was obtained by multiplying the combined weight vector ω calculated by G1 and information entropy with the normalized matrix $F^-$, and the positive ideal plan $z_j^+$ and negative ideal plan $z_j^-$ in the operation plan of the NIMBY facility PPP project to be evaluated were determined as follows:

**Table 4. NIMBY facility PPP project operation mode selection measurement index weight.**

| Destination layer | Criterion layer | Index level | G1 weight | Information entropy weight | Combination weight |
|---|---|---|---|---|---|
| NIMBY facility items PPP are used as the weights of the selected weights of the square weights | Project Features | Project Construction Category C1 | 0.037 | 0.045 | 0.041 |
| | | Financing Scale C2 | 0.059 | 0.037 | 0.047 |
| | | Project market demand C3 | 0.112 | 0.038 | 0.071 |
| | Government ability | Government PPP Inherent Experience C4 | 0.052 | 0.064 | 0.059 |
| | | Government policy leans towards C5 | 0.047 | 0.031 | 0.038 |
| | | Government fiscal capacity C6 | 0.062 | 0.052 | 0.057 |
| | | Government regulatory capability C7 | 0.048 | 0.069 | 0.06 |
| | Project implementation | Financing process complexity C8 | 0.033 | 0.042 | 0.038 |
| | | Technical adaptability C9 | 0.048 | 0.054 | 0.051 |
| | | Complexity of property rights change C10 | 0.051 | 0.063 | 0.058 |
| | Project benefit | Return on investment level C11 | 0.094 | 0.076 | 0.084 |
| | | Charging and pricing mechanismC12 | 0.079 | 0.093 | 0.087 |
| | | Investment payback period C13 | 0.034 | 0.062 | 0.05 |
| | Financing risk | Financial risk C14 | 0.058 | 0.049 | 0.053 |
| | | Construction risk C15 | 0.081 | 0.065 | 0.072 |
| | | Political risk C16 | 0.074 | 0.082 | 0.078 |
| | | Force Majeure Risk C17 | 0.063 | 0.087 | 0.076 |

Subjective weight, objective weight and combined weight are summarized to facilitate comparative analysis.

$z_j^+ = (0.041, 0.047, 0.071, 0.059, 0.038, 0.057, 0.060, 0.038, 0.051, 0.058, 0.084, 0.087, 0.050, 0.053, 0.072, 0.078, 0.076)$;

$z_j^- = (0,0,0,0,0,0,0,0,0,0,0,0,0,0,0,0,0)$

Step 3. According to the formula, the grey correlation degree $Y_i^+$, $Y_i^-$ and Euclidean distance $d_i^+$, $d_i^-$ of the six alternative solutions with positive ideal solutions $z_j^+$ and negative ideal solutions $z_j^-$ are calculated as follows:

$r_j^+ = (0.7827, 1.0000, 0.7852, 0.9437, 0.7969, 0.9669)$

$r_j^- = (0.7323, 0.8171, 0.7214, 0.8439, 0.7833, 1.0000)$

$d_i^+ = (0.8091, 0.9584, 0.7372, 0.7917, 0.7248, 1.0000)$

$d_i^- = (0.6989, 1.0000, 0.6286, 0.9053, 0.6734, 0.9497)$

Step 4. $Y_i^+$, $Y_i^-$, $d_i^+$ and $d_i^-$ are normalized according to the formula, and the relative closeness degree $\pi_i$ of each operation mode is calculated according to the formula. The calculation results of each operation mode are listed in Table 5.

In summary, we can get the comprehensive ranking result of the six operational modes as follows: BOT > BOO > TOT > DBFO > DBO > Equity Transfer. Therefore, it can be concluded that BOT is the best operation mode of the NIMBY facility PPP project, and in practice,

**Table 5. Relative progress of each mode of operation $\pi_i$.**

| The case | DBO | B00 | DBFO | BOT | TOT | stock right transfer |
|---|---|---|---|---|---|---|
| Relative adjacency sort of πi | 0.490109 | 0.529731 | 0.492202 | 0.53062 | 0.493654 | 0.489353 |
| | 5 | 2 | 4 | 1 | 3 | 6 |

The scores and rankings for the six modes are listed.

the NIMBY facility PPP project in Qingdao also adopts such operation mode. The results are consistent with the actual situation, which proves the effectiveness of the model.

In order to verify the rationality and accuracy of the method proposed in this paper, the single TOPSIS method, Grey target model [62] and the matter-element model [63] were respectively used to reorder the operation methods in the calculation examples in this paper, and the results were compared and analyzed. When selecting the evaluator, in order to ensure the reliability of the evaluation information, 8 experts in this field are still selected (Table 6) to evaluate the operating modes of the other 3 methods. After calculation, the results of the four methods are compared, and the results of different methods are listed in Table 7.

As can be seen from Table 7, the single TOPSIS method can not better distinguish the differences and sensitivity of various schemes, leading to deviation in the results. However, the GRA-TOPSIS model can avoid this shortcoming, indicating that this method is more accurate and reasonable. GRA-TOPSIS method makes up for the shortcoming that TOPSIS method can only express the lack of distance and cannot determine the development trend of the scheme. It uses the grey correlation degree between the selected object and the positive and negative ideal solutions to construct the relative closeness degree to select the scheme, thus improving the effectiveness and scientific nature of the selection and making the relative closeness degree of the selection result more accurate. This will help decision makers make more accurate choices. However, GRA-TOPSIS method is not suitable for projects requiring small errors due to its low prediction accuracy.

The grey target model, the matter-element model and the GRA-TOPSIS model have the same calculation results, BOT is the best choice, but compared with the grey target model and the matter-element model, the GRA-TOPSIS model optimizes the calculation method of relative proximity degree, and its algorithm process is simpler, more operable and practical.

Through the above analysis we can see that in the adjacent from the infrastructure PPP project works in the process of selection, large extent the result of the choice of indicators are: C11 investment income level, the C3 project demand, C13 investment payback period, the C4 government PPP inherent factors such as experience, therefore, project earnings, project their

**Table 6. Information of the experts.**

| 1 | University of Cambridge | Professor |
|---|---|---|
| 2 | University of Oxford | Professor |
| 3 | London Engineering Consulting Institute | Professional advisor |
| 4 | America Urban Construction Group | Chief engineer |
| 5 | Beijing City Financing Department | Official |
| 6 | China Construction Eighth Engineering Division Co., LTD | General manager |
| 7 | Morgan Lewis Law Firm | Lawyer |
| 8 | Japan International Engineering Consulting Corporation | Professional advisor |

The occupations and organizations of the eight experts are listed.

**Table 7. Comparison of the results of different methods.**

| Method | Operation mode to sort results | | | | | |
|---|---|---|---|---|---|---|
| TOPSIS method | BOO | BOT | DBFO | TOT | DBO | Stock right transfer |
| Grey target model | BOT | BOO | TOT | DBFO | DBO | Stock right transfer |
| Extended matter-element model | BOT | BOO | TOT | DBFO | DBO | Stock right transfer |
| GRA—TOPSIS | BOT | BOO | TOT | DBFO | DBO | Stock right transfer |

The six models were calculated separately using four different methods and their results were ranked.

own characteristics and the government capability of the adjacent facilities the choice of the PPP project works play a key role. The BOT method for financing the PPP project of NIMBY facilities can greatly relieve the financial pressure on the government, improve the operational efficiency of the PPP project of NIMBY facilities, reduce the financing risk, and thus obtain more comprehensive benefits.

## 6. Conclusion

The construction of NIMBY facilities can effectively provide shared services for the general public, generate corresponding social benefits and meet social needs, but at the same time, it will also produce a certain degree of objective negative externalities, which will cause direct or indirect risks and losses to nearby residents. Due to the asymmetry of costs and the imbalance of benefits and risks, the operation mode selection of NIMBY facility PPP project is of great significance for the sustainable development of the city and the solution of financial problems. This paper constructs a model to select the operation mode of NIMBY facility PPP projects, which is not only of great significance to the sustainable development of the city and the solution of the funding problem, but also plays an important role in selecting the best operation mode of NIMBY facility PPP projects. This paper mainly does the following work: (1) To sort out the operation process and the main classification of PPP projects of NIMBY facilities in China, and build a more perfect evaluation index system to avoid the redundancy of evaluation indexes. (2) The combination of G1 subjective weighting method and information entropy objective weighting method is used to give weight to the index system, which greatly reduces the influence of experts' subjective preferences and ensures the accuracy and rationality of the index weight; The operation mode selection model of NIMBY facility PPP project is established by using TOPSIS method improved by grey correlation theory, which makes the selection process of operation mode of NIMBY facility PPP project more practical and scientific. (3) The effectiveness of this method is verified through the case analysis of Qingdao municipal waste comprehensive treatment PPP project, which is expected to provide reference for the construction and operation of future NIMBY facility PPP projects. The operation mode selection of NIMBY facility PPP project is a hot topic of current research, and has also attracted the attention of researchers. The goals of future research should be aimed into the direction of using the GRA-TOPSIS method for other real problems as well as combining with objective and subjective criteria weighting techniques. Furthermore, one of the goals of future research also lies in expanding GRA-TOPSIS method by using different uncertainty theories. There are many articles analyzing the PPP project of NIMBY facilities, but few literature selecting its operation mode. This paper attempts to contribute to this section.

There are limitations and deficiencies to this study. In the context of the lack of sufficient experience in the NIMBY facility PPP project and the imperfect evaluation index system of operation mode, the data collection should be more accurate and feasible. In the next study, the method of selecting the operation mode model of NIMBY facility PPP projects can be

innovated. If other new methods emerge, the results of the different methods can then be compared.

## Supporting information

**S1 Data. The information entropy data.**
(PDF)

## Acknowledgments

The authors would like to extend their gratitude to the 8 experts for questionnaire respondents for valuable contributions to this research project.

## Author Contributions

**Conceptualization:** Jingqi Zhang.

**Data curation:** Jingqi Zhang.

**Formal analysis:** Jingqi Zhang.

**Funding acquisition:** Jingqi Zhang.

**Investigation:** Hui Zhao, Jingqi Zhang.

**Methodology:** Hui Zhao, Jingqi Zhang.

**Project administration:** Hui Zhao, Jingqi Zhang.

**Resources:** Hui Zhao, Jingqi Zhang.

**Software:** Hui Zhao, Jingqi Zhang.

**Supervision:** Jingqi Zhang.

**Validation:** Yuanyuan Ge.

**Visualization:** Yuanyuan Ge.

**Writing – original draft:** Yuanyuan Ge.

**Writing – review & editing:** Hui Zhao, Yuanyuan Ge.

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
