## [Decision Letter · Decision Letter 0]

6 Jun 2021

PONE-D-21-12807

Operation mode selection of NIMBY facility PPP projects based on combination weighting and GRA-TOPSIS

PLOS ONE

Dear Dr. Zhang,

Thank you for submitting your manuscript to PLOS ONE. After careful consideration, we feel that it has merit but does not fully meet PLOS ONE’s publication criteria as it currently stands. Therefore, we invite you to submit a revised version of the manuscript that addresses the points raised during the review process.

We look forward to receiving your revised manuscript.

Kind regards,

Dragan Pamucar

Academic Editor

PLOS ONE

Journal Requirements:

PLOS requires an ORCID iD for the corresponding author in Editorial Manager on papers submitted after December 6th, 2016. Please ensure that you have an ORCID iD and that it is validated in Editorial Manager. To do this, go to ‘Update my Information’ (in the upper left-hand corner of the main menu), and click on the Fetch/Validate link next to the ORCID field. This will take you to the ORCID site and allow you to create a new iD or authenticate a pre-existing iD in Editorial Manager. Please see the following video for instructions on linking an ORCID iD to your Editorial Manager account: https://www.youtube.com/watch?v=_xcclfuvtxQ

Please include a separate caption for each figure in your manuscript.

Reviewers' comments:

Reviewer's Responses to Questions

**Comments to the Author**

1. Is the manuscript technically sound, and do the data support the conclusions?

Reviewer #1: Partly

Reviewer #2: Partly

2. Has the statistical analysis been performed appropriately and rigorously? 

Reviewer #1: No

Reviewer #2: N/A

3. Have the authors made all data underlying the findings in their manuscript fully available?

Reviewer #1: No

Reviewer #2: Yes

4. Is the manuscript presented in an intelligible fashion and written in standard English?

Reviewer #1: No

Reviewer #2: Yes

5. Review Comments to the Author

Reviewer #1: The authors propose an interesting paper in project management and decision making, which can be viewed in the general framework of modeling. The improvement is suggested in terms of the following comments:

1. Please take care to the formulations. For example, the weight of a method does not sound well. Also, there are too many abbreviations.

2. The abstract should be improved. It needs to cover five main elements: introduction, problem statement, methodology, contributions and results.

3. In introduction section the authors should provide more information about the existing models (GRA-TOPSIS and Entropy) used in the field and their benefits/weaknesses.

4. Why the grey correlation theory has been used in the TOPSIS method? What are the benefits of its implementation in TOPSIS?

5. Recent references in this field of work should be addressed in the overview:

A VIKOR and TOPSIS focused reanalysis of the MADM methods based on logarithmic normalization, Facta Universitatis series: Mechanical Engineering, 18(3), 341-355, 2020.

COPRAS model, Military Technical Courier, 68(1), 28-64, 2020.

Sustainable supplier selection using combined FUCOM - Rough SAW model, Reports in Mechanical Engineering, 1(1), 34-43, 2020.

Tensor product-based model transformation approach to tower crane systems modeling, Asian Journal of Control, DOI: 10.1002/asjc.2494, 2021.

6. You should add step by step calculations in the case study section. You should also follow the steps defined in the preliminaries section.

7. A validation section is missing. How can we assess and interpret the results? Comparison with the existing methods from the literature is missing.

8. The resolution of figures requires improvement.

Reviewer #2: The submitted paper match the aims and scope of PLOS ONE. It represents an application of GRA-TOPSIS tools for evaluation of PPP projects, while G1 and Entropy methods were used for determining the criteria weights in MCMD methodology. The paper would be of interest to the research community in this field of work but it needs substantial improvements in order to be considered for publication in PLOS ONE. I would suggest a series of changes that would improve the paper:

- The introduction section must be improved. Authors should better highlight the objective of their work and what its contributions are in order to close a gap to the existing literature and/or practice. What is the innovative value of the contribution proposed by the authors?

- In the introduction section, the authors should provide more information about existing MCDM models used in the field and their benefits/weaknesses.

- Why the TOPSIS method? Why not RAFSI, MABAC, MAIRCA, VIKOR methods? This should be discussed. The authors need to discuss their contributions compared to those methods in related papers.

- The authors must clearly discuss the significance of the research problem in the first section.

- The authors should explain why they insist on using the G1 subjective weight model. Why not LBWA, FUCOM or BWM models? These models have numerous advantages and the authors should show what the advantages of G1 over LBWA, FUCOM or BWM models are. These methods should be discussed in the paper.

- Why the Entropy method was used (objective weight model)? Why not CRITIC, FANMA, etc?

- The authors should provide more recent references published in the previous two-three years. References published prior to 2017 may be used, but those are generally of less interest. Some recent interesting references from the MCDM field are missing (application of TOPSIS method). I suggest to the authors to read and discuss the below listed references:

A VIKOR and TOPSIS focused reanalysis of the MADM methods based on logarithmic normalization. Facta universitatis series: Mechanical Engineering, 18(3), 341-355.;

A cloud TOPSIS model for green supplier selection. Facta Universitatis, series: Mechanical Engineering. 18(3), pp. 375-397.;

Route planning for hazardous materials transportation: Multicriteria decision making approach. Decision Making: Applications in Management and Engineering, 2(1), 66-85.

A hybridized IT2FS-DEMATEL-AHP-TOPSIS multicriteria decision making approach: Case study of selection and evaluation of criteria for determination of air traffic control radar position. Decision Making: Applications in Management and Engineering, 3(1), 146-164.

- The authors should provide step by step calculations for suggested methodology, especially Comprehensive weight model. The methodology should be explained in details.

- How can a reader judge about the quality of the obtained solutions? Could the results be compared with some existing approaches in literature? The improvement must be discussed. Suggestion: compare the obtained results with results yielded by the classic TOPSIS method and, in doing that, compare the advantages and limitations of your model over the classic TOPSIS method.

- A validation section is missing.

- In the conclusion section, the authors need to demonstrate the impact and insights of the research. The authors need to clearly provide several solid future research directions and their research contributions. The limitations of the model should be added.

6. PLOS authors have the option to publish the peer review history of their article (what does this mean?). If published, this will include your full peer review and any attached files.

Reviewer #1: No

Reviewer #2: No

---

## [Author Response · Author response to Decision Letter 0]

15 Jun 2021

First of all, thank you very much for your encouraging and inspiring feedback on my work and for your constructive and helpful comments that have definitely improved the paper a great deal. I had studied all of your comments carefully and tried to incorporate all of them into the current version of this article.

---

## [Decision Letter · Decision Letter 1]

17 Jun 2021

PONE-D-21-12807R1

Operation mode selection of NIMBY facility Public Private Partnership

PLOS ONE

Dear Dr. Zhang,

Thank you for submitting your manuscript to PLOS ONE. After careful consideration, we feel that it has merit but does not fully meet PLOS ONE’s publication criteria as it currently stands. Therefore, we invite you to submit a revised version of the manuscript that addresses the points raised during the review process.

We look forward to receiving your revised manuscript.

Kind regards,

Dragan Pamucar

Academic Editor

PLOS ONE

Journal Requirements:

Reviewers' comments:

Reviewer's Responses to Questions

**Comments to the Author**

1. If the authors have adequately addressed your comments raised in a previous round of review and you feel that this manuscript is now acceptable for publication, you may indicate that here to bypass the “Comments to the Author” section, enter your conflict of interest statement in the “Confidential to Editor” section, and submit your "Accept" recommendation.

Reviewer #1: All comments have been addressed

Reviewer #2: (No Response)

2. Is the manuscript technically sound, and do the data support the conclusions?

Reviewer #1: Yes

Reviewer #2: Yes

3. Has the statistical analysis been performed appropriately and rigorously? 

Reviewer #1: Yes

Reviewer #2: Yes

4. Have the authors made all data underlying the findings in their manuscript fully available?

Reviewer #1: No

Reviewer #2: Yes

5. Is the manuscript presented in an intelligible fashion and written in standard English?

Reviewer #1: Yes

Reviewer #2: Yes

6. Review Comments to the Author

Reviewer #1: The revised paper is improved. It is principally acceptable for publishing, but the authors should check all the references and provide full bibliographic data for each reference.

Reviewer #2: The authors have improved the paper significantly by properly addressing all the comments. There is only one technical aspect that must be additionally taken care of. Namely, a number of references are incomplete, because they are missing the journal titles and further data. For instance:

- in reference 15 the missing journal title is "Reports in Mechanical Engineering", volume and issue missing as well;

- in references 14 and 17, the missing journal title is "Facta Universitatis series Mechanical Engineering", also volume and issue number are missing;

- In reference 19, the missing journal title is "Decision Making: Applications in Management and Engineering", volume and issue missing as well;

- in reference 20, the missing journal title is "Journal of Finance Research", also volume, issue and page numbers are missing here.

Those were only examples. Please, check all the references and make sure all of them contain complete data.

Provided this comment has been properly addressed, the paper would be acceptable for publishing.

7. PLOS authors have the option to publish the peer review history of their article (what does this mean?). If published, this will include your full peer review and any attached files.

Reviewer #1: No

Reviewer #2: No

---

## [Author Response · Author response to Decision Letter 1]

20 Jun 2021

Response to Reviewer 1’s Comments

Thank you very much for your valuable comments and suggestions. We attach great importance to your suggestion on revising the reference. We have checked all the references and highlighted the changes in yellow. Thank you again for your help.

Response to Reviewer 2’s Comments

First of all, thank you very much for your encouraging and inspiring feedback on my work and for your constructive and helpful comments that have definitely improved the paper a great deal. We attach great importance to your suggestion on revising the reference. We have checked all the references and highlighted the changes in yellow. Thank you again for your help.

---

## [Decision Letter · Decision Letter 2]

21 Jun 2021

Operation mode selection of NIMBY facility Public Private Partnership

PONE-D-21-12807R2

Dear Dr. Zhang,

We’re pleased to inform you that your manuscript has been judged scientifically suitable for publication and will be formally accepted for publication once it meets all outstanding technical requirements.

Kind regards,

Dragan Pamucar

Academic Editor

PLOS ONE

Additional Editor Comments (optional):

Reviewers' comments:

Reviewer's Responses to Questions

**Comments to the Author**

1. If the authors have adequately addressed your comments raised in a previous round of review and you feel that this manuscript is now acceptable for publication, you may indicate that here to bypass the “Comments to the Author” section, enter your conflict of interest statement in the “Confidential to Editor” section, and submit your "Accept" recommendation.

Reviewer #1: All comments have been addressed

Reviewer #2: All comments have been addressed

2. Is the manuscript technically sound, and do the data support the conclusions?

Reviewer #1: Yes

Reviewer #2: Yes

3. Has the statistical analysis been performed appropriately and rigorously? 

Reviewer #1: Yes

Reviewer #2: Yes

4. Have the authors made all data underlying the findings in their manuscript fully available?

Reviewer #1: No

Reviewer #2: Yes

5. Is the manuscript presented in an intelligible fashion and written in standard English?

Reviewer #1: Yes

Reviewer #2: Yes

6. Review Comments to the Author

Reviewer #1: The authors did a great job in the revision of this very good manuscript. Their efforts are appreciated.

The responses to my comments are well documented and supported by relevant information.

Moreover, all these responses are reflected in the improvement of the manuscript. Therefore, the revised manuscript is correct and technically sound.

I especially appreciate the validation. This revised manuscript contains a sound theory and an illustrative validation, which accompanies and also supports the theoretical claims.

Concluding, I agree again with the publication of this manuscript. This manuscript will certainly continue authors' other past well-acknowledged manuscripts in the growing field of the manuscript.

I estimate that the manuscript will have a high impact for the readers of this highly ranked journal.

Reviewer #2: All comments have been addressed properly. Therefore, the paper is recommended for publishing as it is.

7. PLOS authors have the option to publish the peer review history of their article (what does this mean?). If published, this will include your full peer review and any attached files.

Reviewer #1: No

Reviewer #2: No

---

## [Editor Report · Acceptance letter]

29 Jun 2021

PONE-D-21-12807R2 

Operation mode selection of NIMBY facility Public Private Partnership projects 

Dear Dr. Zhang:

I'm pleased to inform you that your manuscript has been deemed suitable for publication in PLOS ONE. Congratulations! Your manuscript is now with our production department. 

Kind regards, 

on behalf of

Dr. Dragan Pamucar 

Academic Editor

PLOS ONE